# Salvadoran Celastraceae Species as a Source of Antikinetoplastid Quinonemethide Triterpenoids

**DOI:** 10.3390/plants13030360

**Published:** 2024-01-25

**Authors:** Marvin J. Núñez, Morena L. Martínez, Ulises G. Castillo, Karla Carolina Flores, Jenny Menjívar, Atteneri López-Arencibia, Carlos J. Bethencourt-Estrella, Ignacio A. Jiménez, José E. Piñero, Jacob Lorenzo-Morales, Isabel L. Bazzocchi

**Affiliations:** 1Laboratorio de Investigación en Productos Naturales (LIPN), Facultad de Química y Farmacia, Universidad de El Salvador, Final Ave. Mártires Estudiantes del 30 de Julio, San Salvador 01101, El Salvador; marvin.nunez@ues.edu.sv (M.J.N.); morena.martinez@ues.edu.sv (M.L.M.); ulises.guardado@ues.edu.sv (U.G.C.); kcflor30@gmail.com (K.C.F.); 2Museo de Historia Natural de El Salvador, Ministerio de Cultura, Final Calle Los Viveros, Col. Nicaragua, San Salvador 01101, El Salvador; jenny79menjivar@gmail.com; 3Instituto Universitario de Enfermedades Tropicales y Salud Pública de Canarias, Universidad de La Laguna, 38200 La Laguna, Tenerife, Spain; atlopez@ull.edu.es (A.L.-A.); cbethene@ull.edu.es (C.J.B.-E.); jpinero@ull.es (J.E.P.); jmlorenz@ull.es (J.L.-M.); 4Departamento de Obstetricia y Ginecología, Pediatría, Medicina Preventiva y Salud Pública, Toxicología, Medicina Legal y Forense y Parasitología, Universidad de La Laguna, 38200 La Laguna, Tenerife, Spain; 5Centro de Investigación Biomédica en Red de Enfermedades Infecciosas (CIBERINFEC), Instituto de Salud Carlos III, 28220 Madrid, Spain; 6Departamento de Química Orgánica, Instituto Universitario de Bio-Orgánica Antonio González, Universidad de La Laguna, Avenida Astrofísico Francisco Sánchez 2, 38206 La Laguna, Tenerife, Spain; ignadiaz@ull.es

**Keywords:** *Trypanosoma cruzi*, *Leishmania* spp., Celastraceae, phytochemical profile, antikinetoplastid

## Abstract

Chagas disease and leishmaniasis are among the most widespread neglected tropical diseases, and their current therapies have limited efficacy and several toxic side effects. The present study reports the chemical and antikinetoplastid profiles of extracts from five Salvadoran Celastraceae species against the *Trypanosoma cruzi* epimastigotes stage and *Leishmania amazonensis* and *Leishmania donovani* promastigote forms. The phytochemical profile evinced the presence of flavonoids, tannins, sterols, and triterpenes as the main components in all plant species, whereas quinonemethide triterpenoids (QMTs) were restricted to the root bark of the studied species. Antikinetoplastid evaluation highlights the root bark extracts from *Zinowewia integerrima*, *Maytenus segoviarum,* and *Quetzalia ilicina* as the most promising ones, exhibiting higher potency against *T. cruzi* (IC_50_ 0.71–1.58 µg/mL) and *L. amazonensis* (IC_50_ 0.38–2.05 µg/mL) than the reference drugs, benznidazole (IC_50_ 1.81 µg/mL) and miltefosine (IC_50_ 2.64 µg/mL), respectively. This potent activity was connected with an excellent selectivity index on the murine macrophage J774A.1 cell line. These findings reinforce the potential of QMTs as antikinetoplastid agents for the development of innovative phytopharmaceuticals and the plant species under study as a source of these promising lead compounds.

## 1. Introduction

According to the World Health Organization, Chagas disease and leishmaniasis stand as two of the foremost neglected tropical diseases (NTDs), arising from various species of intracellular kinetoplastid parasites, with the potential to lead to epidemic outbreaks [1]. These NTDs disproportionately impact impoverished and marginalized communities and are prevalent in tropical and subtropical areas across Africa, the Americas, Asia, and Oceania [2].

Chagas disease is caused by infection with *Trypanosoma cruzi*, a flagellated protozoan transmitted by insects belonging to the *Triatominae* subfamily. This NTD exerts a significant impact on Latin America, with instances of congenital and transfusion-based transmissions altering its global epidemiology. The disease manifests in two clinical forms: the acute and chronic phases. Approximately 75% of infected individuals remain asymptomatic throughout their lives, while 25% of cases lead to heart, digestive, and nervous system complications. The annual incidence of Chagas disease reaches 28,000 cases in the Americas region, affecting an estimated 6 to 8 million people and resulting in approximately 12,000 deaths each year [3]. 

Furthermore, leishmaniasis encompasses a spectrum of maladies induced by *Leishmania* protozoa and transmitted by female sandflies from the *Phlebotomus* and *Lutzomyia* genera. The annual death worldwide toll due to leishmaniasis falls within the range of 20,000 to 30,000, with approximately 350 million individuals being susceptible to infection [4]. Cutaneous leishmaniasis (CL) ranks as the most prevalent manifestation of this infection, causing ulcerative lesions that often lead to lifelong scarring [5]. In recent years, the incidence of leishmaniasis as an opportunistic disease has increased, primarily due to heightened immunosuppression resulting from chronic conditions, such as HIV infection [6].

Between 2018 and 2020, a study in El Salvador, covering 107 households, revealed a 34.4% infection rate among approximately 1500 examined triatomines, vectors of the Chagas disease, and no less than 10% of them having the potential to transmit *T. cruzi* to humans [7]. In 2018, a surveillance study in western El Salvador found that the seropositivity rate for pediatric Chagas disease in the Sonsonate Department was 2.3%, surpassing the reported national adult rate of 1.7% [8]. In 2022, a maternal surveillance study of 198 pregnant women identified a *T. cruzi* positivity rate of 6% using serology or molecular diagnosis. Furthermore, half of the newborns of *T. cruzi*-positive mothers required treatment for neonatal complications [9]. Furthermore, from 2001 to 2021, El Salvador’s official records indicate a total of 834 cases of CL and visceral leishmaniasis (VL), marking a substantial increase of 28.2% since 2020 [10]. Notably, in the Central America region, atypical cutaneous leishmaniasis (ACL) predominates. In 2017, 43 cases of ACL were reported, all of them in children under 5 years of age [11], whereas in 2021, among the 50 reported cases, 46% were affected children under 10 years of age [10], signifying a worrying change in the affected demographic.

Chagas disease can be treated with benznidazole or nifurtimox, both with almost 100% efficacy at the onset of the acute phase and in congenital transmission cases. However, treatment effectiveness decreases the longer a person has been infected [12]. 

Moreover, the primary treatment for leishmaniasis involves pentavalent antimonials, while second-line treatments for resistant cases encompass amphotericin B and its liposomal variant, miltefosine, pentamidine, azole drugs, and paromomycin [13]. Despite advancements in chemotherapy, the treatment of these infections faces challenges such as varying sensitivity among species, toxicity, administration methods, resistance, adverse effects, and cost-related issues [2]. Consequently, due to the multiple limitations associated with current chemotherapy and the absence of an effective human vaccine, there is an increasing urgency to develop novel drugs that are both effective and safe and that can be made more readily accessible.

Plant-derived products represent a promising source of potent and structurally diverse scaffolds for improving drug development to treat these diseases [14]. In this regard, species of the Celastraceae family have extensively been used in traditional medicine, and therefore, they stand out as promising plant sources of bioactive components [15]. The Celastraceae family includes around 94 genera and 1410 species of trees, shrubs, or climbers distributed in tropical and sub-tropical regions of the world [16]. Twenty-two Celastraceae species have been identified so far in El Salvador, mainly from the *Wimmeria*, *Zinoweiwia*, *Maytenus*, *Euonymus,* and *Celastrus* genera [17]. 

In a continuous research program for new antikinetoplastid agents and encouraged by our previous results on *Maytenus chiapensis* [18], the present work deals with the phytochemical profile and evaluation of five Salvadoran Celastraceae species against the epimastigote stage of *Trypanosoma cruzi* and promastigote forms of *Leishmania amazonensis* and *Leishmania donovani*. It is worth noting that the study of *Quetzalia ilicina*, *Wimmeria cyclocarpa*, and *Euonymus enantiophyllus* is reported herein for the first time. The results underline the root bark extracts as a promising source of new antikinetoplastid agents, exhibiting potent effects on *T. cruzi* (IC_50_ 0.71–3.45 µg/mL) and *L. amazonensis* (IC_50_ 0.38–6.96 µg/mL) coupled with good selectivity.

## 2. Results and Discussion

In the present work, five Salvadoran Celastraceae species, *Maytenus segoviarum*, *Q. ilicina*, *Zinowiewia integerrima*, *W. cyclocarpa,* and *E. enantiophyllus,* were studied in the search for antiketenoplastid natural sources. The details of their collection, including voucher specimen, place of collection, and geographic coordinates, are summarized in Table 1. Thus, the different parts of the plants, i.e., aerial (leaves, branches, and fruits) and root bark parts, were studied separately. The fruits of *Z. integerrima* and *E. enantiophyllus* could not be included in this study since they were not available during the collection stage of these species.

### 2.1. Phytochemical Analysis

The dried powdered material from the five species under study was subjected to different extraction processes. Thus, leaves (25 g) or branches (25 g) parts were extracted with ethanol and further fractionated by liquid–liquid partition to afford the dichloromethane and *n*-butanol fractions (Section 3.3 in Materials and Methods). Moreover, dried powdered fruit or root bark parts were extracted to obtain the hexanes/Et_2_O (1:1), methanolic, and acetonic extracts. The yields obtained for each extract or fraction are detailed in Table 2.

A preliminary phytochemical profile assay of the 34 extracts/fractions from aerial (leaves, branches, fruits) and root bark parts of the Celastraceae species under study was carried out by a Thin Layer Chromatography (TLC) method, using silica gel 60F_254_ as stationary phase and mixtures of organic solvents as mobile phase. The detection of secondary metabolites was followed by UV light at 254 nm or 365 nm, using selective detection reagents, and comparison with standards [19]. The results indicated the presence of saponins, flavonoids, tannins, alkaloids, coumarins, sterols, triterpenes, and QMTs in all the tested species, whereas none of them exhibited cardiotonic glycosides, anthraquinones, or sesquiterpene lactones. Moreover, flavonoids, tannins, sterols, and triterpenes were found in all studied organs (root bark, leaves, branches, and fruits), whereas the fruits of *Q. ilicina* as well as the root bark of *E. enantiophyllus* were the only ones containing coumarins. All the analyzed root bark extracts contain saponins, flavonoids, tannins, alkaloids, sterols, triterpenes, and QMTs (Table 3, Appendix A). 

Saponins were detected in all the species under study, which is in agreement with previous phytochemical studies in Celastraceae species [20,21,22]. Flavonoids were detected in all the analyzed organs, in agreement with studies on two Brazilian species, *Maytenus aquifolium* and *Maytenus ilicifolia* [23], used as anti-ulcer phytopharmaceuticals. Investigations on species from *Crossopetalum* [24], *Celastrus* [25,26], and *Euonymus* [27] genera also reported flavonoids showing antiproliferative, antioxidant, and neuroprotective properties. Regarding the analysis of tannins, those were detected in all organs and all species under study. Tannins, among the most abundant phenolic secondary metabolites in nature, exert protective functions against crop pests and infections and environmental factors such as UV radiation, heat, or drought [28]. Moreover, the presence of tannins has been evidenced in Celastraceae species extensively used in South American traditional medicine [22,29,30,31]. Sesquiterpene pyridine alkaloids are a type of macrolides restricted to Celastraceae species, and particularly abundant in *Maytenus*, *Celastrus*, *Euonymus,* and *Tripterygium* genera. Sesquiterpene alkaloids isolated from a species of the Salvadoran flora, *M. chiapensis*, have been reported to possess insect antifeedant activity against *Spodoptera littoralis* [32]. In addition, this type of metabolites has been reported from the stems of *M. segoviarum*, and the leaves of *Celastrus vulcanicola* [33]. In the present study, alkaloids were detected in *M. segoviarum* and *E. enantiophyllus* and for the first time in species of *Quetzalia*, *Zinowiewia*, and *Wimmeria* genera. 

Triterpenes and sterols are a group of abundant metabolites in the Celastraceae species. In this study, all species and organs analyzed presented this type of secondary metabolites. Previous studies reported pentacyclic and tetracyclic triterpenes from Salvadoran Celastraceae species, including *M. chiapensis* [34], *Crossopetalum uragoga* [35], *C. xylocarpa*, [36] and *C. vulcanicola* [37], and those isolated from *M. chiapensis* [34] and *C. vulcanicola* [37] represented unprecedented tetracyclic *D:B*-friedobaccharane, and montecrinane skeletons, respectively. Coumarins represent a class of aromatic secondary metabolites derived from the shikimic acid pathway, which have a wide variety of biological activities [38]. Coumarins are rare in Celastraceae species since only four of them have been reported from *Gymnosporia senegalensis* var. *spinosa* [39], *Euonymus hamiltonianus* [40], the Brazilian species, *Maytenus guianesis* [20], and recently, in *Monteverdia communis* [41]. The results in the present work are in accordance with those previous reported, since coumarins were only detected in the fruits of *Q. ilicina* and root bark of *E. enantiophyllus*. Cardiotonic glycosides are restricted almost exclusively to *Elaeodendron* [42] and *Crossopetalum* [24] species, whereas anthraquinones and sesquiterpene lactones have not been reported from Celastracea species. All root bark extracts analyzed presented QMTs, a relatively limited group of biologically active compounds restricted to the Celastraceae species and are regarded as chemotaxonomic markers for this plant family [43]. QTMs have been reported exclusively in the root bark of species from *Maytenus* [18,20], *Euonymus* [44], and *Celastrus* genera [45]. 

### 2.2. Antikinetoplastid Activity

In the search for new plant sources of antikinetoplastid agents, the 34 extracts/fractions of the five species under study were evaluated in vitro on epimastigote stage of *T. cruzi* and promastigote forms of *L. amazonensis* and *L. donovani*. The results are expressed as the inhibitory concentration of the sample that eliminates 50% of the population of parasites (IC_50_). Cytotoxicity, expressed as the cytotoxic concentration in 50% of the cell population (CC_50_), was also determined in a normal eukaryotic cell line (murine macrophages J7741A.1) to test selectivity. The selectivity index (SI) was calculated from the CC_50_ divided by the IC_50_. Benznidazole and miltefosine were used as reference drugs against *Trypanosoma cruzi* and *Leishmania* spp., respectively.

#### 2.2.1. Trypanocidal Activity

The results of the in vitro assay against *T. cruzi* epimastigote stage revealed that among the 34 extracts/fractions tested, seven root bark extracts from three species, *Z. integerrima, M. segoviarum,* and *Q. ilicina* exhibited a potent anti-trypanosomal activity (Table 3). The hexanes:Et_2_O (1:1) extracts of *Z. integerrima* (IC_50_ = 0.71 µg/mL) and *M. segoviarum* (IC_50_ = 1.36 µg/mL), and the acetone extracts from *Z. integerrima* (IC_50_ = 0.75 µg/mL) and *Q. ilicina* (IC_50_ = 1.58 µg/mL) showed higher potency than the widely known reference drug, benznidazole, used as a positive control (IC_50_ = 1.81 µg/mL) (Table 4). 

Moreover, the methanolic extracts of *Z. integerrima* (IC_50_ = 2.87 µg/mL) and *Q. ilicina* (IC_50_ = 3.06 µg/mL), and the hexanes:Et_2_O (1:1) extract of *Q. ilicina* (IC_50_ = 3.45 µg/mL) exhibited also a significant anti-trypanosomal activity. In addition, all of them showed an excellent SI, with values ranging from 58.0 to 99.4 versus that of benznidazole (SI 57.5) (Table 4). In particular, highlight the hexanes:Et_2_O (1:1) extract of *Z. integerrima*, exhibiting 2.5-fold greater antikinetoplastid effectiveness than the reference drug coupled also with an excellent SI (99.4) on the mammalian cell line. Taking into account that QTMs are the most characteristic components of the root bark of Celastraceae species, these results pointed out this type of metabolites as responsible for the anti-trypanosomal activity of the species under study. This statement is in agreement with previous works reporting that QTMs isolated from *Cheiloclinium cognatum* [46], and *M. chiapensis* [18] showed inhibition of the growth of *T. cruzi* epimastigotes. Moreover, Goijman and coworkers [47], reported that tingenone, an abundant QMT in Celastraceae species, inhibited *T. cruzi* growth by DNA double-strand intercalation.

#### 2.2.2. Leishmanicidal Activity

The extracts/fractions of the five Celastraceae species under study were tested against *L. amazonensis* and *L. donovani* promastigote forms. The results indicated that *L. amazonensis* is much more sensitive to the assayed samples than *L. donovani*. In fact, extracts from four species, *Z. integerrima*, *M. segoviarum*, *Q. ilicina*, and *E. enantiophyllus* showed remarkable activity against *L. amazonensis*, with IC_50_s ranging from 0.38 to 2.05 µg/mL, even lower than the reference drug, miltefosine, used as a positive control (IC_50_ = 2.64 µg/mL) (Table 5).

Among the studied species, highlight *Z. integerrima*, since all organic extracts (acetonic, hexanes/Et_2_O (1:1), and methanolic extracts) from the root bark evaluated showed potent activity (IC_50_s 0.38–1.13 µg/mL), being around 7-, 4.5-, and 2.3-fold, respectively, higher than the reference drug. Moreover, these activities were coupled with an excellent selectivity index in murine macrophages (SI ranging from 119.6 to 177.0), even higher than this for miltelfosine (SI = 11.1) (Table 4). Besides, three root bark extracts, the hexanes/Et_2_O (1:1) extracts of *Q. ilicina* (IC_50_ = 3.06 µg/mL) and *W. cyclocarpa* (IC_50_ = 6.96 µg/mL) and the acetone extract of *E. enantiophyllus* (IC_50_ = 4.58 µg/mL) (Table 4), showed significant activity. Nevertheless, in the case of *L. donovani*, only the acetone extract from the root bark of *Q. ilicina* showed some degree of activity (IC_50_ = 5.42 µg/mL, SI = 20.5) (see Appendix A). Certainly, the presence of QMTs in the root bark extracts underlines this type of metabolite as one of the bioactive components in these plant species and points them out as sources of these promising antikinetoplastid agents. Furthermore, prior studies exploring the mechanism of action of QMTs in *L. amazonensis* revealed mitochondrial damage, characterized by a loss of membrane potential without ATP-level alterations, suggesting that apoptosis may be the underlying physiological mechanism of cell death in the parasite [18]. 

The present work investigates the chemical and antikinetoplastid profiles of extracts from five Salvadoran species of Celastraceae to address the limited efficacy and toxic side effects associated with current therapies against Chagas disease and leishmaniasis. The study reveals a novel approach by identifying quinonemethide triterpenoids (QMTs) in the root bark of specific species and associating them with potent antikinetoplastid activity against *Trypanosoma cruzi* and *Leishmania* species. The comprehensive phytochemical profile shows the presence of flavonoids, tannins, sterols, and triterpenes as major constituents in all the plants studied, providing a detailed insight into their chemical composition.

Although the present study demonstrates promising in vitro results, it has limitations that deserve consideration, such as the lack of in vivo data, suggesting the need for further research to validate the efficacy and safety of potential treatments derived from these plants. In addition, the specificity of the geographical locations for certain Celastraceae species raises questions about the scalability of the results. Nevertheless, the study provides a valuable starting point by offering a comparative analysis with reference drugs and selectivity values.

The prospects of this research focus on the possible development of drugs against both tropical diseases. To this end, the QMTs identified present promising candidates for possible isolation, paving the way for in-depth studies of the mechanism of action. Future in vivo experiments are crucial to validating the efficacy and safety of the isolated extracts or compounds in a more physiologically relevant context. Exploring synergies with existing antiparasitic drugs could also improve treatment efficacy and reduce the risk of resistance development. Ultimately, successful in vivo trials could position these plant extracts as innovative phytopharmaceuticals for the treatment of Chagas disease and leishmaniasis.

## 3. Materials and Methods

### 3.1. Chemicals and Reagents

All solvents and reagents used in the phytochemical profile analysis were of analytical grade (ACS) and purchased from Sigma-Aldrich (Saint Louis, MO, USA). Pristimerin and tingenone, isolated from *Maytenus chiapensis* (Celastraceae) [18], and 2,3-epoxyjuanislamin, isolated from *Calea urticifolia* (Asteraceae) [48], were used as comparison standards after their NMR characterization. Sephadex LH-20 used for column chromatography (CC) was supplied by Pharmacia Biotech. Silica gel 60 (particle sizes 15–40 and 63–200 μm, Macherey-Nagel) and silica gel 60F_254_ used for CC and analytical and preparative thin layer chromatography, respectively, were purchased from Panreac (Barcelona, Spain). Benznidazole from Sigma-Aldrich (St Louis, MO, USA) and miltefosine from Æterna Zentaris (Charleston, SC, USA) were used as reference drugs for trypanocidal and leishmanicidal activities, respectively. 

### 3.2. Plant Material

Plant species were collected in the Santa Ana Department (El Salvador). The details of their collection, including the vernacular name, voucher specimen, place of collection, and geographic coordinates of the five plant species, are summarized in Table 1. All species were identified by the Curator of the Herbarium at the Museum of Natural History of El Salvador (MUHNES), Jenny Elizabeth Menjívar Cruz. A voucher specimen of each plant was deposited at the MUHNES.

### 3.3. Preparation of Plant Extracts

The plant material was dried in a programmable air-circulating oven (Thermo Scientific Heratherm OMH400) at 40 °C for 72 h and proceeded to grind (particle size 1–2 mm). The dried powdered material from the five species under study was subjected to different extraction processes. Leaves (25 g) or branches (25 g) parts were extracted by magnetic stirrer ultrasonic (VWR, model 97043-988, operating frequency at 35 kHz) with 250 mL of 95° ethanol for 1.5 h at 25 °C, and then filtered. The ethanolic extracts were concentrated under reduced pressure at 40 °C, providing 2.0 g and 1.7 g of crude extract, respectively, and further fractionated by liquid–liquid partition following a modification of Kupchan’s method [49], using dichloromethane-*n*-butanol instead of chloroform-*n*-butanol as solvents. Thus, the extract was suspended in distilled water (100 mL) and successively extracted with dichloromethane (50 mL) and *n*-butanol (50 mL). The organic phases were dried over Na_2_SO_4_, filtered, and brought to dryness on a rotary evaporator to afford the dichloromethane (F-D) and *n*-butanol (F-B) fractions. Moreover, dried powdered fruit or root bark parts were extracted by dissolving 10 g of plant material with 100 mL of hexanes–Et_2_O (1:1), methanol, or acetone, and then sonicated for 1.5 h at 25 °C (VWR, model 97043-988, operating frequency at 35 kHz). The extracts were concentrated under reduced pressure at 40 °C to obtain crude residues, hexanes/Et_2_O (1:1) extracts, and acetonic extracts. All extracts and fractions were assayed for antikinetoplastid activity.

### 3.4. Phytochemical Profile Analysis by TLC

Phytochemical tests to identify saponins, cardiac glycosides, flavonoids, anthraquinones, alkaloids, sesquiterpene lactones, coumarins, and triterpenes were carried out by thin-layer chromatography (TLC) as described by Mejía et al. [19]. A 10 mg/mL methanolic solution was prepared from each extract or fraction, and 5 to 10 μL was seeded on the chromatographic plate. To identify tannins, the extracts and fractions were subjected to TLC, using ethyl acetate/methanol/water (8:1:1) as the eluent and tannic acid as a positive control. The TLC was sprayed with iron trichloride 10%, and dark-blue spots were taken as positive evidence [50]. Sterol identification was carried out using hexanes/ethyl acetate (1:1) as eluent, and after development, TLC was visualized using Liebermann–Burchard’s reagent, whereas β-sitosterol was used as a positive control. Positive evidence for sterols begins with purplish spots and progresses until a dark green color (blue, green, pink, brown, yellow, and purple spots) on the TLC [51]. For QTM identification, 60 mg of each root bark organic extract was fractionated using a Sephadex column LH-20 and 100 mL *of* hexanes/chloroform/methanol (2:1:1) as the eluent. Four resulting fractions were subjected to TLC, using hexanes/ethyl acetate (1:1) as the eluent and oleum as the dyeing reagent. The well-known QMTs pristimerin and tingenone [18] were used as positive controls, and the evidence was yellow and orange spots on the TLC that turned purple after being sprayed with oleum and subsequent heating.

### 3.5. Antikinetoplastid Activity

#### 3.5.1. Parasite Cultures

Experiments were conducted using the following strains: *Leishmania amazonensis* (MHOM/BR/77/LTB0016), *Leishmania donovani* (MHOM/IN/90/GE1F8R), and *Trypanosoma cruzi* (Y strain). Promastigotes of *Leishmania* strains were cultured in Schneider’s medium (Sigma-Aldrich, Madrid, Spain) supplemented with 10% fetal bovine serum (Biowest, VWR) at 26 °C. They were grown until reaching the log phase before being used in further experiments. To perform the experiments, the parasites were also cultured in RPMI 1640 medium (Gibco^®^), with or without phenol red. Epimastigotes of *T. cruzi* were cultured in Liver Infusion Tryptose (LIT) medium supplemented with 10% fetal bovine serum (Biowest, VWR) at 26 °C [18].

#### 3.5.2. In Vitro Anti-Trypanosomal and Anti-Leishmanicidal Activity Assay 

The extracts and fractions were tested in vitro in the epimastigote stage of *T. cruzi* by a colorimetric assay based on alamarBlue reagent, as previously outlined [18]. Briefly, the tested samples were serially diluted in 100 µL of RPMI-1640 medium without phenol red and supplemented with 10% SBF in 96-well plates. Parasites in the log growth phase were counted, diluted (10^5^/well), and added to these wells. Subsequently, 10% alamarBlue reagent was added to the plates and incubated at 26 °C. After 72 h, the plates were analyzed using an EnSpire Multimode Plate Reader by relative fluorescence unit measurement. 

The extracts and fractions were tested against the promastigote stage of both *L. amazonensis* and *L. donovani* using a colorimetric assay based on alamarBlue^®^ reagent (Invitrogen, Life Technologies, Madrid, Spain) as previously described [18]. Briefly, the tested samples were serially diluted in 100 μL of either RPMI 1640 medium without phenol red or LIT medium, depending on the parasite, in 96-well plates. Parasites in log phase growth were counted, diluted (10^5^/well), and added to these wells. Finally, 10% of alamarBlue^®^ was added to the plates, and these were incubated at 26 °C. After 72 h, the plates were analyzed using an EnSpire^®^ Multimode Plate Reader (Perkin Elmer, Madrid, Spain) to measure the fluorescence of alamarBlue^®^ (excitation 530 nm, emission 590 nm).

In both in vitro assays, the percentage of inhibition and 50% inhibitory concentrations (IC_50_) were calculated by no linear regression analysis with 95% confidence limits using SigmaPlot 12.0 statistical analysis software (Systat Software). All experiments were performed three times each in duplicate, and the mean values were calculated.

#### 3.5.3. Cytotoxicity Assays

Murine macrophages (J774A.1 cell line) were cultured in RPMI-1640 medium, counted, and subsequently seeded in 96-well plates (10^5^ cells/mL). The test sample was diluted in culture medium and added to a total volume of 100 μL in each well, as previously detailed [18]. As a negative control, cells were incubated with culture medium alone. Finally, 10 μL of alamarBlue was added to each well, and the plate was incubated for 24 h at 37 °C in a 5% CO_2_ atmosphere. The plates were analyzed using an EnSpire microplate reader. The cytotoxic concentration (CC_50_) was calculated using Sigma Plot 12.0 statistical analysis software. The selectivity index was the ratio between the CC_50_ value on murine cells and the IC_50_ value on parasites. 

#### 3.5.4. Statistical Analysis 

All assays were carried out in triplicate on separate days. Sigma Plot 12.0 (Systat Software, Grafiti LLC, Palo Alto, CA, USA) statistical analysis software was used to calculate IC_50_ and CC_50_ values by non-linear regression with 95% confidence limits. The results are expressed as the mean of the three replicates ± standard deviation. Analysis of variance was performed by one-way ANOVA, and differences of *p* < 0.05 were considered statistically significant.

## 4. Conclusions

The current study reports our efforts to search for new alternatives to current treatments for kinetoplastid diseases. In this study, the root bark extracts of three Celastraceae species, *Z. integerrima*, *M. segoviarum*, and *Q. ilicina,* were successfully identified as effective plant species against *T. cruzi* and *L. amazonensis,* exhibiting potent activities coupled with an excellent selectivity index, better than that of reference drugs currently in clinical use. Taking into consideration the presence of QMTs in the root bark of these species, it could be pointed out that these metabolites are the bioactive components, reinforcing previous studies. These results demonstrate that these species could be applied as an extract or enriched fraction in a simple extraction process step, which gives additional value to the potential uses of the Celastraceae plant species under study. Furthermore, they are a rich resource of QMTs, chemotaxonomic markers of the Celastraceae species, highlighting the therapeutic potential of these promising candidates as future drugs for Chagas disease and leishmaniasis treatments. 

Based on the robust in vitro anti-parasitic activity and favorable toxicity profile demonstrated by these plant species, we are poised to embark on a trajectory toward the advancement of phytopharmaceuticals. Overall, this work shows a practical strategy for the discovery of antikinetoplastid agents from natural sources and provides insights regarding QMTs as promising scaffolds for the development of novel agents meaningful to kinetoplastid therapy in the future.

## Figures and Tables

**Table 1 plants-13-00360-t001:** Details of the collected Salvadoran Celastraceae species under study.

Plant Specie/Vernacular Name	Voucher Number/Date	Collection Place ^a^	Geographic Coordinates ^b^
*Maytenus segoviarum* Standl. & L.O. Williams/“Jocotillo”	J. Menjivar et al. 4001/ May 2019	Cantón El Limo, Metapán	14°24′6″ N, 89°25′7″ W/1040 m.a.s.l.
*Quetzalia ilicina* (Standl. & Steyerm.) Lundell/“Palo de palomo”	J. Menjívar et al. 3970/March 2019	Parque Nacional Montecristo, Metapán	14°24′55″ N, 89°21′34″ W/2180 m.a.s.l.
*Zinowiewia integerrima* (Turcz.) Turcz/“Barreto”	J. Menjívar et al. 3971/March 2019	Parque Nacional Montecristo, Metapán	14°23′52″ N, 89°21′42″ W/1840 m.a.s.l.
*Wimmeria cyclocarpa* Radlk./“Palo rojizo”	J. Menjívar et al. 4130/May 2019	Parque Nacional Montecristo, Metapán	14°23′26″ N, 89°22′45″ W/1670 m.a.s.l.
*Euonymus enantiophyllus* (Donn. Sm.) Lundell/“Palo bonito”	J. Menjívar et al. 4216/ November 2019	Parque Nacional Montecristo, Metapán	14°25′4″ N, 89°21′18″ W/2300 m.a.s.l.

^a^ All species were collected in Santa Ana, El Salvador. ^b^ Geographic coordinates: latitude, longitude, and elevation.

**Table 2 plants-13-00360-t002:** Details of the yields (% *w*/*w*) obtained for extracts and fractionations from the Celastraceae species.

	Plant Part	R. Bark E/M	R. Bark E/A	R. Bark E/H-E	Leaves E/Et	Leaves F/D	Leaves F/B	Branches E/Et	Branches F/D	Branches F/B	Fruits E/M	Fruits E/A	Fruits E/H-E
Species													
*M. segoviarum*		32.5	9.3	3.6	16.1	1.7	6.2	10.5	1.5	3.0	9.0	2.0	1.1
*Q. ilicina*		15.6	7.1	3.7	7.6	4.3	2.8	6.8	1.6	0.4	11.3	1.9	0.8
*Z. integerrima*		10.6	0.3	5.7	12.2	3.4	1.0	6.0	0.9	0.8	-	-	-
*W. cyclocarpa*		6.9	10.6	5.9	9.6	0.8	3.8	4.4	2.3	1.5	10.7	3.1	1.3
*E.enantiophyllus*		21.0	8.2	5.0	4.4	2.4	0.2	4.5	1.0	1.5	10.9	0.8	1.1

R. bark = root bark; E = extract; F = fraction; M = methanol; A = acetone; H-E = hexanes/EtO_2_O (1:1); DCM = dichloromethane; BuOH = *n*-butanol; Et = ethanol.

**Table 3 plants-13-00360-t003:** Preliminary phytochemical profile of screened Celastraceae species.

		Secondary Metabolite	Saponins	Cardiac glycosides	Flavonoids	Anthraquinone sglycoside	Tannins	Alkaloids	Sesquiterpene lactones	Coumarins	Sterols	Triterpenes	QMTs
Species	Plant Part												
*Maytenus segoviarum*	Root bark		+	−	+	−	+	+	−	−	+	+	+
	Leaves		+	−	+	−	+	+	−	−	+	+	−
	Branches		+	−	+	−	+	+	−	−	+	+	−
	Fruits		−	−	+	−	+	−	−	−	+	+	−
*Quetzalia ilicina*	Root bark		+	−	+	−	+	+	−	−	+	+	+
	Leaves		−	−	+	−	+	+	−	−	+	+	−
	Branches		−	−	+	−	+	+	−	−	+	+	−
	Fruits		−	−	+	−	+	−	−	+	+	+	−
*Zinowiewia integerrima*	Root bark		+	−	+	−	+	+	−	−	+	+	+
	Leaves		+	−	+	−	+	+	−	−	+	+	−
	Branches		+	−	+	−	+	+	−	−	+	+	−
*Wimmeria cyclocarpa*	Root bark		+	−	+	−	+	+	−	−	+	+	+
	Leaves		+	−	+	−	+	−	−	−	+	+	−
	Branches		−	−	+	−	+	−	−	−	+	+	−
	Fruits		−	−	+	−	+	+	−	−	+	+	−
*Euonymus enantiophyllus*	Root bark		+	−	+	−	+	+	−	+	+	+	+
	Leaves		−	−	+	−	+	+	−	−	+	+	−
	Branches		+	−	+	−	+	+	−	−	+	+	−
	Fruits		−	−	+	−	+	−	−	−	+	+	−

+: Present; −: Absent; QMTs: quinonemethide triterpenoids.

**Table 4 plants-13-00360-t004:** Trypanocidal activity (IC_50_) and selectivity index (SI) of root bark extracts of selected Celastraceae species against epimastigote stage of *T. cruzi* and cytotoxicity against eukaryotic cells.

Specie	Extract or Drug	*T. cruzi* IC_50_ (µg/mL) ^a^	Murine Macrophages CC_50_ (µg/mL) ^b^	SI ^c^
*Z. integerrima*	Hexanes:Et_2_O (1:1)	0.71 ± 0.04	70.59 ± 7.94	99.4
	Acetone	0.75 ± 0.07	57.81 ± 1.86	77.1
	MeOH	2.87 ± 0.39	>200	69.7
*M. segoviarum*	Hexanes:Et_2_O (1:1)	1.36 ± 0.17	103.58 ± 2.78	76.2
*Q. ilicina*	Acetone	1.58 ± 0.03	111.33 ± 11.84	70.5
	MeOH	3.06 ± 0.68	>200	63.4
	Hexanes:Et_2_O (1:1)	3.45 ± 0.35	>200	58.0
Control ^d^	Benznidazole	1.81 ± 0.50	104.1 ± 1.04	57.5

^a^ IC_50_: concentrations able to inhibit 50% of parasites after 72 h, expressed as µg/mL ± standard deviation (SD). ^b^ CC_50_ concentration able to inhibit 50% of murine macrophages after 24 h, expressed as µg/mL ± standard deviation (SD). ^c^ SI: selectivity index (CC_50_/IC_50_). ^d^ Control: Benznidazole was used as the positive control against *T. cruzi*.

**Table 5 plants-13-00360-t005:** Leishmanicidal activity (IC_50_) and selectivity index (SI) of root bark extracts of selected Celastraceae species against promastigote stage of *L. amazonensis* and cytotoxicity against eukaryotic cells.

Specie	Extract	*L. amazonensis* IC_50_ (µg/mL) ^a^	Murine Macrophages CC_50_ (µg/mL) ^b^	SI ^c^
*Z. integerrima*	Acetone	0.38 ± 0.08	57.81 ± 1.86	152.1
	Hexanes:Et_2_O (1:1)	0.59 ± 0.05	70.59 ± 7.94	119.6
	MeOH	1.13 ± 0.07	>200	177.0
*M. segoviarum*	Hexanes:Et_2_O (1:1)	0.85 ± 0.10	103.58 ± 2.78	121.9
*Q. ilicina*	Acetone	1.34 ± 0.22	111.33 ± 11.84	83.1
	MeOH	1.40 ± 0.21	>200	142.9
	Hexanes:Et_2_O (1:1)	3.06 ± 0.77	>200	65.4
*E. enantiophyllus*	Hexanes:Et_2_O (1:1)	2.05 ± 0.10	>200	97.6
	Acetone	4.58 ± 0.05	>200	43.7
*W. cyclocarpa*	Hexanes:Et_2_O (1:1)	6.96 ± 0.88	79.33 ± 11,3	11.4
Control ^d^	Miltefosine	2.64 ± 0.10	29.42 ± 3.06	11.1

^a^ IC_50_: concentrations able to inhibit 50% of parasites after 72 h, expressed as µg/mL ± standard deviation (SD). ^b^ CC_50_ concentration able to inhibit 50% of murine macrophages after 24 h, expressed as µg/mL ± standard deviation (SD).^c^ SI: electivity index (CC_50_/IC_50_). ^d^ Control: Miltefosine was used as the positive control against *L. amazonensis*.

## Data Availability

The data generated and analyzed during this study are included in this article.

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
