# Peer review of "Salvadoran Celastraceae Species as a Source of Antikinetoplastid Quinonemethide Triterpenoids"

_plants, 2024, doi:10.3390/plants13030360_

Round 1
Reviewer 1 Report
Comments and Suggestions for Authors
Leishmaniasis and Chagas disease are two of the most common neglected tropical diseases; current treatments have a number of harmful side effects as well as limited efficacy. The present study describes the chemical and antinetoplastid profiles of extracts of five species of Celastraceae found in El Salvador, tested against the promastigote form of Leishmania donovani and Amazonensis and the epimastigote stage of Trypanosoma cruzi. All plant species were found to include flavonoids, tannins, sterols and triterpenes as primary constituents, according to the phytochemical profile. The root bark extracts of Zinowewewia integerrima, Maytenus segoviarum and Quetzalia ilicina are the most promising according to the antinetoplastid evaluation; they show greater potency against T. cruzi and L. amazonensis than the reference drugs miltefosine and benznidazole, respectively. This strong action is linked to exceptional selectivity. The excellent selectivity index (SI) on the murine macrophage cell line J774A.1 was linked to this potent action. These results support the use of QMTs as antineoplastic agents in the creation of new phytochemicals, as well as the potential of the plant species studied to provide these intriguing guide compounds.
The manuscript is written in good English, and the study described above in its entirety is interesting and worthy of publication. I have no major comments to make, the only thing I do not understand is why the authors omitted to make the 'Discussion' section. I know it is present in the Plants template, why did you omit it? If there is no reasonable reason, I would ask the authors to edit this section comparing their results with those of other authors.
Author Response
RESPONSES TO REVIEWER 1
Request 1. I have no major comments to make, the only thing I do not understand is why the authors omitted to make the 'Discussion' section. I know it is present in the Plants template, why did you omit it? If there is no reasonable reason, I would ask the authors to edit this section comparing their results with those of other authors.
Answer. Taking in consideration the Guide for authors (Discussion: This section may be combined with Results.) we have included the Discussion together with Results, and in accordance references to other papers, and comparative data from other publications are included in Results.
In the revised manuscript, we have been corrected the head of the Section Results by Results and Discussion.
Reviewer 2 Report
Comments and Suggestions for Authors
The present article highlights the Salvadoran Celastraceae species as a source of antikinetoplastid quinonemethide triterpenoids. The topic is relevant, but some shortcomings identified in both content and form need to be addressed based on the specific recommendations below:
Abbreviations are explained when they first appear in the abstract or main text and help to make the text easier to read and convey information more effectively. Once an abbreviation has been established and explained, it will be used throughout the manuscript, except in the abstract, where it should be treated separately. Please review the entire manuscript and explain abbreviations used directly without the initial long form (e.g., abstract - no SI abbreviation is needed as it does not appear a second time in the abstract, etc.).
Enhance the aim of the paper by emphasizing the novelty of the topic and the potential contributions to the scientific community.
The results section cannot have bibliographic resources as it only presents the results obtained by the authors and nothing else.
References to other papers, comparative data from other publications are included in the separate discussion section as a separate chapter (see the template provided by the journal).
It is advisable to provide more information on the determined concentrations, statistical studies and it would be much better if figures from at least the TLC assessment were provided.
In the last paragraph of the Discussion section, it is advisable to detail the strengths, but especially the limitations of your study and to what extent these could be addressed for future research directions.
Auto-citations should be rechecked and reconsidered because not all of them find their place in the current manuscript.
Author Response
RESPONSES TO REVIEWER 2
Remark 1. Abbreviations are explained when they first appear in the abstract or main text and help to make the text easier to read and convey information more effectively. Once an abbreviation has been established and explained, it will be used throughout the manuscript, except in the abstract, where it should be treated separately. Please review the entire manuscript and explain abbreviations used directly without the initial long form (e.g., abstract - no SI abbreviation is needed as it does not appear a second time in the abstract, etc.).
Answer. The reviewer is right, and the used abbreviations have been revised throughout the manuscript, including SI abbreviation in the abstract.
Remark 2. Enhance the aim of the paper by emphasizing the novelty of the topic and the potential contributions to the scientific community.
Answer: This point has been corrected in the revised manuscript, and a paragraph has been included in the Conclusion section.
“Overall, this work showed a practical strategy for the discovery of antikinetoplastid agents from natural sources and provides insights regarding QMTs as promising scaffolds for the development of novel agents, meaningful to kinetoplastid therapy in the future.”
Remark 3. The results section cannot have bibliographic resources as it only presents the results obtained by the authors and nothing else.
Answer: As the reviewer pointed out the results section cannot have bibliographic resources, however, we have included the Discussion together with Results, and in accordance references to other papers, comparative data from other publications are included in its Section. In the revised manuscript, we have been corrected the head of the Section Results by Results and Discussion.
Remark 4. References to other papers, comparative data from other publications are included in the separate discussion section as a separate chapter (see the template provided by the journal).
Answer: Following the Guide for authors we combined Results and Discussion
Remark 5. It is advisable to provide more information on the determined concentrations, statistical studies and it would be much better if figures from at least the TLC assessment were provided.
Answer: As requested by the Reviewer more information have been included in the revised manuscript. Thus, a paragraph on the concentrations used in the TLC plate for the phytochemical analysis was included in the Section: 3.4 Phytochemical profile analysis by TLC in Materials and Methods.
A section with the details of the statistical analysis is included in Material and Methos in the current version of the manuscript.
“3.5.4. Statistical Analysis. All assays were carried out in triplicate on separate days. Sigma Plot 12.0 (Systat Software) statistical analysis software was used to calculate IC50 and CC50 values by non-linear regression with 95 % confidence limits. The results are expressed as the mean of the three replicates ± standard deviation. Analysis of variance was performed by one-way ANOVA, and differences of p < 0.05 were considered statistically significant.”
In addition, a Figure with TLC images was included in the Supporting Information of the revised manuscript, as Figure S1: Selected Thin Layer Chromatography (TLC) of phytochemical analysis.
Remark 6. In the last paragraph of the Discussion section, it is advisable to detail the strengths, but especially the limitations of your study, and to what extent these could be addressed for future research directions.
Answer. This point has been corrected in the revised manuscript, and a paragraph has been included at the end of Results and Discussion section.
“The present work investigates the chemical and antikinetoplastid profiles of extracts from five Salvadorian species of Celastraceae to address the limited efficacy and toxic side effects associated with current therapies against Chagas disease and leishmaniasis. The study reveals a novel approach by identifying quinonemethide triterpenoids (QMTs) in the root bark of specific species and associating them with potent antikinetoplastid activity against Trypanosoma cruzi and Leishmania species. The comprehensive phytochemical profile shows the presence of flavonoids, tannins, sterols, and triterpenes as major constituents in all the plants studied, providing a detailed insight into their chemical composition.
Although the present study demonstrates promising in vitro results, it has limitations that deserve consideration, such as the lack of in vivo data suggesting the need for further research to validate the efficacy and safety of potential treatments derived from these plants. In addition, the specificity of the geographical locations for certain Celastraceae species raises questions about the scalability of the results. Nevertheless, the study provides a valuable starting point by offering a comparative analysis with reference drugs and selectivity values.
The prospects of this research focus on the possible development of drugs against both tropical diseases. To this end, the QMTs identified present promising candidates for possible isolation, paving the way for in-depth studies of the mechanism of action. Future in vivo experiments are crucial to validate the efficacy and safety of the isolated extracts or compounds in a more physiologically relevant context. Exploring synergies with existing antiparasitic drugs could also improve treatment efficacy and reduce the risk of resistance development. Ultimately, successful in vivo trials could position these plant extracts as innovative phytopharmaceuticals for the treatment of Chagas disease and leishmaniasis.”
Remark 7. Auto-citations should be rechecked and reconsidered because not all of them find their place in the current manuscript.
Answer: This point has been corrected in the revised version of the manuscript.
Reviewer 3 Report
Comments and Suggestions for Authors
Dear authors.
Please find below the observations and comments made on your manuscript. You should update the manuscript accordingly as indicated in the observations and comments.
1. Lines 60-61: Is that the number of deaths in El Salvador, Central America, the Americas or the world? You must indicate.
2. At the beginning you should enter the full name of the plant species, not abbreviated.
3. What were the criteria for selecting the 5 plant species, not clear in the document. Must include entnobotanical, ethnomedical or ethnopharmacological information; or is it by the principle of chance or serendipity.
4. Q. ilicina, W. cyclocarpa, and E. enantiophyllus are reported for the first time in this paper, so under what or what guidelines were these species selected?
5. I can't find the discussion section in the manuscript, only the results section.
6. Can you relate the phytochemical content to the biological activities evaluated (Antikinetoplastid, Trypanocidal, and Leishmanicidal activities)? In other reports, some of the secondary metabolites found in this study show these activities.
7. Can the particle size of the product be indicated by the grinding of the plant species?
8. You should include in the manuscript (text or table) the yields obtained from extraction and fractionation.
9. Include in the paper the comparison between the modification made to the method and the Kupchan method.
10. You should indicate/mention why you used the extraction solvents dichloromethane and butanol for the fractionation, and not others. Because you then mention that you used hexane, acetone and methanol. For direct and quick comparison purposes it would be appropriate to use the same solvents for each species and for each plant organ.
11. For the TLC tests, what were the concentrations of the samples and in which solvent? What was the volume applied/seeded on the chromatoplate?
12. The description for the tests for Saponins, Cardiac glycosides, Flavonoids, Anthraquinones glycoside, Alkaloids, Sesquiterpene lactones, Coumarins, Triterpenes does not appear in the methodology. You should open the methodology indicating the step-by-step for each of the TLC tests. Remember that open methods (methodology) are easily referenced in other articles!
13. At the end of the discussion section you should include a paragraph or several paragraphs indicating the strengths and weaknesses of the study conducted.
14. Question: What was done in each of the 6 institutions that are related to the authors of this paper?
Regards
Reviewer
Author Response
RESPONSES TO REVIEWER 3
Remark 1. Lines 60-61: Is that the number of deaths in El Salvador, Central America, the Americas, or the world? You must indicate.
Answer. This point has been corrected in the revised manuscript (refers to worldwide).
Remark 2. At the beginning you should enter the full name of the plant species, not abbreviated.
Answer. This point has been corrected in the revised manuscript.
Remark 3. What were the criteria for selecting the 5 plant species, not clear in the document. Must include entnobotanical, ethnomedical or ethnopharmacological information; or is it by the principle of chance or serendipity.
Answer. The argument for selecting the 5 plant species under study was the promising results obtained in our previous work on Maytenus chiapensis, collected in El Salvador, against T. cruzi and L. amazonensis (Nuñez et al., Antimicrob. Agents Chemother., 2021). We think it could be interesting to study other species of Salvadoran Celastraceae, as we describe in this manuscript. We carried out an exhaustive search for the species included in the manuscript and they do not have ethnobotanical uses in El Salvador, Central America, and Mexico.
Remark 4. Q. ilicina, W. cyclocarpa, and E. enantiophyllus are reported for the first time in this paper, so under what or what guidelines were these species selected?
Answer. These Celastraceae species have not been previously studied, either from a phytochemical or pharmacological point of view. So, we think their study for the first time is an interesting contribution to the scientific community.
Remark 5. I can't find the discussion section in the manuscript, only the results section.
Answer. Taking in consideration the Guide for authors (Discussion: This section may be combined with Results.) we have included the Discussion together with Results, and in accordance references to other papers, comparative data from other publications are included in Results.
In the revised manuscript, we have been corrected the head of the Section Results by Results and Discussion.
Remark 6. Can you relate the phytochemical content to the biological activities evaluated (Antikinetoplastid, Trypanocidal, and Leishmanicidal activities)? In other reports, some of the secondary metabolites found in this study show these activities.
Answer. In the present work, we have analysed the phytochemical profile of extracts from different parts of the five Salvadoran Celastraceae species (Table 2), and all the organic extracts and fractions were assayed against Trypanosoma cruzi, Leishmania amazonensis, and Leishmania donovani (Table S1). The root bark extracts were the most promising ones, exhibiting high potency against T. cruzi and L. amazonensis. Taking in account that the quinonemethide triterpenoids QMTs are restricted to the main metabolites in the root barks, we could suggest these metabolites are the responsible for activity. In fact, previous report highlights these type of metabolites as promising anti-parasitic agents. Thus, the present work reinforces this previous work.
Remark 7. Can the particle size of the product be indicated by the grinding of the plant species?
Answer. The particle size (1-2 mm) of the grinding plant materials has been indicated in the revised manuscript following the reviewer’s comment in Materials and Methods (Section 3.3 Preparation of Plant Extracts)
Remark 8. You should include in the manuscript (text or table) the yields obtained from extraction and fractionation.
Answer. A Table (Table 2) indicating the yields obtained from extraction and fractionation has been included in the text in the revised version of the manuscript.
Remark 9. Include in the paper the comparison between the modification made to the method and the Kupchan method.
Answer. As the reviewer suggests, the modification of Kupchan’s method has been indicated in the revised manuscript (Section 3.3 Preparation of Plant Extracts), in addition to a reference by Kupchan [56].
Remark 10. You should indicate/mention why you used the extraction solvents dichloromethane and butanol for the fractionation, and not others. Because you then mention that you used hexane, acetone, and methanol. For direct and quick comparison purposes it would be appropriate to use the same solvents for each species and for each plant organ.
Answer. Dichloromethane and butanol were used as extraction solvents, as part of a specific technique of our research laboratory to isolate secondary metabolites. The same solvents are not used for each organ, because according to our previous experiences, a better yield of desired molecules is obtained, using different solvents depending on the plant organ to be extracted.
Remark 11. For the TLC tests, what were the concentrations of the samples, and in which solvent? What was the volume applied/seeded on the chromatoplate?
Answer. This point has been corrected, including a paragraph in the Section 3.4 Phytochemical profile analysis by TLC (A 10 mg/ml methanolic solution was prepared from each extract or fraction and 5 to 10 μL was seeded on the chromatographic plate).
Remark 12. The description for the tests for Saponins, Cardiac glycosides, Flavonoids, Anthraquinones glycoside, Alkaloids, Sesquiterpene lactones, Coumarins, Triterpenes does not appear in the methodology. You should open the methodology indicating the step-by-step for each of the TLC tests. Remember that open methods (methodology) are easily referenced in other articles!
Answer. The phytochemical tests carried out by TLC in this research are routine tests, which are generally referenced in manuscripts regarding phytochemistry articles or books. We have included information on phytochemical tests by TLC in three different sections of the manuscript (2.1, 3.1, and 3.4) with three references from us (ref 18 ref 19 and, 55), and two from other research groups (ref 56 and ref 57).
Remark 13. At the end of the discussion section, you should include a paragraph or several paragraphs indicating the strengths and weaknesses of the study conducted.
Answer. This point has been corrected in the revised manuscript, and a paragraph has been included at the end of the Results and Discussion section.
Remark 14. Question: What was done in each of the 6 institutions that are related to the authors of this paper?
Answer. The phytochemical profile of the species under study was carried out at:
1 Laboratorio de Investigación en Productos Naturales, Facultad de Química y Farmacia, Universidad de El Salvador.
The collection and identification of the plant species were carried out by the Curator of the Herbarium at the Museum of Natural History of El Salvador (MUHNES), Jenny Elizabeth Menjívar Cruz from the Institution:
2 Museo de Historia Natural de El Salvador, Ministerio de Cultura.
All the biological assays were carried out by co-authors affiliated at different Institutions:
3 Instituto Universitario de Enfermedades Tropicales y Salud Pública de Canarias, Universidad de La Laguna,
4 Departamento de Obstetricia y Ginecología, Pediatría, Medicina Preventiva y Salud Pública, Toxicología, Medicina Legal y Forense y Parasitología, Universidad de La Laguna
5 Centro de Investigación Biomédica en Red de Enfermedades Infecciosas (CIBERINFEC), Instituto de Salud Carlos III, 28220 Madrid, Spain
The conceptualization, the writing-review and funding acquisition were carried out by co-authors at:
6 Instituto Universitario de Bio-Orgánica Antonio González, Departamento de Química Orgánica, Universidad de La Laguna
Round 2
Reviewer 2 Report
Comments and Suggestions for Authors
The authors have significantly improved the manuscript based on the suggestions received.
Author Response
REVIWER 2’ COMMENTS (Round 2)
Remark: The authors have significantly improved the manuscript based on the suggestions received.
Answer: Thanks for your consideration.
Reviewer 3 Report
Comments and Suggestions for Authors
Dear authors
For practical purposes and quickly I request to inform in which part of the document where the changes were made indicating the line number(s) and page number(s). I suggest using a different colour for the letters
Reviewer
Author Response
REVIEWER 3’ COMMENTS (Round 2)
Remark: For practical purposes and quickly I request to inform in which part of the document where the changes were made indicating the line number(s) and page number(s). I suggest using a different colour for the letters.
Answer: A copy of the revised manuscript highlighted in yellow with changes to the text was included in the resubmission.